# First Measurement of Ambient Air Quality on the Rural Lower Eastern Shore of Maryland

**Bernice Bediako and Deborah G. Sauder ***

Department of Natural Sciences, University of Maryland Eastern Shore, Princess Anne, MD 21853, USA; bbediako@umes.edu
* Correspondence: dgsauder@umes.edu

**Abstract:** Concerns about atmospheric ammonia have been expressed recently by some on the Lower Eastern Shore (LES) of Maryland, which lies between the Chesapeake Bay and the Atlantic Ocean on the Delmarva peninsula. Agriculture, seafood and tourism are responsible for a significant fraction of the economic activity on the LES. The USDA 2017 census reported there were ~100 Concentrated Animal Feeding Operations (CAFOs) raising nearly 63 M chickens per year across Somerset and Worcester Counties. We report air quality data collected from sites near Princess Anne, Somerset County, and near Pocomoke City, Worcester County, to address air quality concerns by examining the influence of chicken farms on ammonia in ambient air on the LES. Within a two-mile radius of the Worcester County site, CAFO operations house ~1.6 million birds. The Princess Anne site is comparable to the Pocomoke City site in agricultural use and population demographics but has only a few chicken houses within two miles. The first 33 months of LES ammonia data are presented, and their significance is discussed relative to other ammonia studies. The 33-month average concentration of ammonia in Pocomoke was 10.3 ± 0.08 ppb, more than double that in Princess Anne, which was 4.7 ± 0.04 ppb.

**Keywords:** poultry CAFO; ammonia; agricultural impact on air quality

## 1. Introduction

Measurements made by the Lower Eastern Shore Ambient Air Quality Monitoring Project [1] are presented and discussed in this paper. The project collects scientific data to improve understanding of the impact of CAFOs on ambient air quality on the LES of Maryland. Atmospheric concentrations of ammonia, and a variety of N-containing compounds, as well as simultaneous weather conditions, are measured at two rural sites, one which has a minimal number of CAFOs within a two-mile radius and the second a rural site with similar demographics but surrounded by a substantially higher number of CAFOs. The first site is near Princess Anne, Somerset County, and the second is near Pocomoke City, Worcester County. The results of this project are contrasted with existing Eastern Shore data collected by MDE at Horn Point in Dorchester County and at Old Town in Baltimore City [1].

Project partners include the Maryland Department of the Environment (MDE), the Keith Campbell Foundation for the Environment (The Campbell Foundation), the Delmarva Chicken Association (DCA) (formerly Delmarva Poultry Industry, Inc., DPI) and the University of Maryland Eastern Shore (UMES). The Campbell Foundation and DCA provided funding to purchase the equipment (Teledyne-API, San Diego, USA) and monitoring stations used to collect the data. MDE and UMES maintained all site operations and were responsible for all data collection, data quality assurance and data-sharing activities. Continuous measurement of ambient air quality for this project began on 1 April 2020. This paper reports and discusses the results with regards to ammonia.

The earliest study of ammonia emissions from chicken houses on the Delmarva peninsula was published by Siefert et al. in 2004 [2]. They reported ammonia measurements

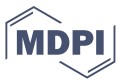

over 6–12 h over seven days during May, June and July 2002. The upwind and downwind sampling of a complex of chicken houses near Princess Anne utilized Ogawa passive samplers, and measurements were performed when winds on site were from the SSW, aligned with the long axis of the chicken houses. All measurements were taken within 200 m of the chicken houses. This data allowed the authors to use inverse modeling to determine the ammonia source strength from the houses and compare the model results with emission factors available at the time. The paper noted the need for gas-phase ammonia measurements to improve understanding of the impact of CAFOs on air quality in the region.

The results of the Southeastern Aerosol Research and Characterization study (SEARCH) were published by Blanchard et al. in 2013 [3]. Data collected from 1999 to 2012 from eight sites in Mississippi, Alabama, Georgia and Florida were considered, along with data from other EPA networks and meteorological data, to provide a comprehensive picture of urban and non-urban sites across the region. The ammonia measurements reported showed highest concentrations (mean value of 2.67 ppbv from 2004 to 2012) at Yorkville, GA (YRK), described as a non-urban site ~65 km northwest of Atlanta. Ammonia concentrations at YRK were considered influenced by power plants located ~25 km and ~85 km away, by pollution from Atlanta and by the presence of poultry farms. Birmingham, AL (BHM), and Jefferson Street in midtown Atlanta, GA (JST), were the only other sites, both urban, with ammonia means that exceeded 1.0 ppbv.

More recently, Baker et al. [4] reported the results of American Meteorological Society (AMS)/U.S. EPA Regulatory Model (AERMOD) simulations using Carnegie-Mellon University (CMU) emission factors to estimate the nitrogen deposition to the Maryland Eastern Shore and the Chesapeake Bay from CAFOs on the Maryland Eastern Shore. To calibrate their model, they took measurements of atmospheric ammonia concentrations at 23 sites along the Bay over two two-week periods in September and October 2017. Using passive CEH Adapted Low-Cost Passive High Absorption (ALPHA) samplers, they reported ammonia concentrations from 0.66 ppb to 4.75 ppb, with an average value of 1.80 ppb and a standard deviation of 1.22 ppb. Two samplers were placed in areas with high CAFO density, where ammonia concentrations of 1.44 (±1 sd 0.12 ppb) and 1.26 ppb (±1 sd 0.12 ppb) were reported, respectively, for the first 2-week period. During the second sampling session, the observed ammonia concentrations were 1.65 (±1 sd 0.012 ppb) and 1.63 ppb (±1 sd 0.012 ppb), at the two sites with high CAFO density. Across all the sampling sites, less spatial variability was observed during the second 2-week period, with a mean concentration of 1.48 ppb and standard deviation of 0.32 ppb.

The authors reported uncertainty in the results of their simulations because local weather conditions were not measured. The National Weather Service station at the Salisbury-Ocean City Wicomico County Regional Airport was assumed to describe weather conditions at the sampling sites. They further highlighted the value of measuring concentrations of ammonia near ground-level to elucidate the influence of ammonia scavenging and particulate matter formation in the model. A further complication acknowledged was the assumption that the entire Eastern Shore was treated as agricultural land.

Most recently on the Lower Eastern Shore of Maryland (LES), a citizen group has undertaken fence line studies which measure atmospheric ammonia levels near a few CAFOs [5]. They have released annual averaged data and distance from the nearest CAFO at four sites. The results show that ammonia concentrations decline as the distance at which the monitors are installed increases from the CAFO, as expected from point sources.

In this paper, we have focused on ammonia because it is the basis of citizen health concerns. We provide 33 months of ammonia concentrations measured at 1 h resolution and confirmed by 2-week average passive measurements. The data improve the temporal resolution in comparison to other measurements made on the LES. In addition, we record weather conditions measured at our sampling sites to remove the limitations of using centralized weather data, which may or may not reflect conditions at dispersed regional sites.

The data reported here were collected within a framework designed to determine the range of ambient ammonia concentrations to which individuals on the Lower Eastern Shore of Maryland are typically exposed. To bracket the range of ambient concentrations, we identified two sites as similar as possible, but varying in one characteristic- one with high CAFO density within two miles, and one with low CAFO density. The measurements reported here report the levels of atmospheric ammonia from two characteristic environments, rather than focusing on emissions from specific CAFOs, or depending on simulations to predict ammonia concentrations. In this way, these data provide a complement to previous studies which considered the issue of atmospheric ammonia from other perspectives.

Ammonia is not a criteria pollutant and therefore is not subject to regular monitoring in pursuit of the US Environmental Protection Agency's (EPA) National Ambient Air Quality Standard (NAAQS). Nation-wide measurement of atmospheric ammonia concentrations began in 2007, when the Ammonia Monitoring Network (AMoN) system was established as a special study by the EPA and the NADP. Permanent status for the AMoN system as an official NADP network was granted in October 2010. Data accumulated by many AMoN sites, including the two discussed in this paper, are archived and available at the NADP web site [6]. The hourly data recorded in this work for $NH_3$, $PM_{2.5}$ and $PM_{10}$ concentrations and weather conditions are posted on the Maryland Department of the Environment's Lower Eastern Shore Ambient Air Quality Monitoring Project web site [1].

Since ammonia is not a criteria pollutant, there are no regulations or outdoor air quality standards for ammonia; however, there are rules on emissions from specific categories of industrial emitters which require them to report releases, including releases of ammonia [7]. To put the results reported here into some regulatory context, we highlight some existing ammonia regulations. The EPA has set the human "no-observed-adverse-effect level" (NOAEL) as 2.3 mg/m$^3$ [8]. Under atmospheric standard conditions, 25 °C and a pressure of 1 atmosphere, this is equivalent to 3.3 ppm $NH_3$ [9]. The American Conference of Governmental Industrial Hygienists (ACGIH) has established a Threshold Limit Value (TLV) for ammonia of 25 ppm for an 8 h exposure period. TLVs refer to the airborne concentrations of a substance that represents conditions to which nearly all workers may be exposed without adverse health effects. ACGIH has also established a TLV of 35 ppm for a 1 h exposure limit [10]. The Occupational Health and Safety Administration (OSHA) has set a Permissible Exposure Limit for ammonia of 50 ppm averaged over an eight-hour workday. This is the standard that must be met in every workplace [11]. The National Institute for Occupational Safety and Health (NIOSH), an arm of the Centers for Disease Control (CDC), provides a Recommended Exposure Limit (REL) for ammonia of 25 ppm averaged over an eight-hour workday. NIOSH also says that there should be a Short-Term Exposure Limit (STEL) for ammonia of 35 ppm during any 15 min period in the day [12]. The NIOSH recommendations are currently under reconsideration. The Maryland Department of the Environment Air Quality Division has established its Ammonia Air Toxics Screening Level at 250 ppb, averaged over 8 h, and 350 ppb for 1 h of exposure, a factor of 100 times lower than the ACGIH or NIOSH levels, and close to one tenth the EPA NOAEL recommendation [1].

CAFOs are agricultural operations where animals are kept and raised in confined situations for a total of 45 days or more in a 12-month period. On the shore, typical CAFOs house six or seven flocks per year. The presence of large numbers of animals in a locality has been documented to contribute to degraded air quality, both in the US and around the world. In 2008, the National Research Council (NRC) Acute Exposure Guideline Levels for Selected Airborne Chemicals: Volume 6 [13] supported calls for further studies of the agriculturally based emissions of ammonia. The factors influencing ammonia emissions from natural and agricultural sources are numerous and subject to substantial variability based on local conditions [14,15].

The monitoring of CAFOs by the EPA began in 2005 [16]. Since 2006, The National Air Emissions Monitoring Study (NAEMS) has examined the emission of atmospheric pollutants from livestock facilities. These efforts are continuing today [17]. In addition to

the AMoN program, other experimental measurements of ambient atmospheric ammonia using passive samplers have been reported from locations around the world since the 1990s [18]. Bittman et al. reported the results of a one-year series of experiments in which passive samplers were deployed across the Lower Fraser Valley in BC, Canada, to study the ammonia concentrations associated with the repopulation of poultry following an outbreak of avian flu in 2004 [19]. Their measurements show that ambient ammonia concentrations increased from 4.9 $\mu g/m^3$ to 13.4 $\mu g/m^3$ in an agriculturally intensive sector as the poultry population increased from 0 to 228,000 chickens per 4 km $\times$ 4 km grid. The contributions of other livestock, manure storage and spreading, and non-farm activities are also modeled in their report. Roadman et al. [20] published a study of the efficacy of Ogawa passive samplers in determining ammonia concentrations in and around chicken houses in Delaware, in 2003. The data provided show a decline in $NH_3$ concentrations from 330–4400 $\mu g$ $NH_3$-N/m$^3$ to a background agricultural level of <25 $\mu g$ $NH_3$-N/m$^3$ as the measuring devices are moved 20 m away from a CAFO.

Felix et al. [21] used ALPHA passive samplers to determine the $^{15}N/^{14}N$ isotopic ratios at nine sites across the US from Texas in the west, Michigan in the north and South Carolina in the southeast over 12 months from July 2009 to June 2010. The studies provided information about the sources of ammonia (agricultural/natural biological processes vs. combustion processes) at the sites and were compared with collocated AMoN results. They found that the ammonia concentrations averaged 3.5 $\mu g/m^3$ annually and were higher in the summer than in the fall/winter. (Spring data for the site were not available.) In addition, they found that the $\delta^{15}N/NH_3$ value was always negative, averaging $-17.1\%$ for the NC site reflecting the importance of hog and chicken farms in determining the air quality around the NC site. Other isotope experiments were reported by Walters et al. [22], who quantified the importance of vehicle emissions to ammonia concentrations measured in Providence, RI, an urban center in the northeastern USA. Their experiments simultaneously collected reactive gases and $PM_{2.5}$. A series of coated glass honeycomb denuders and a downstream filter pack housed in a ChemComb Speciation Cartridge were used. Lab analysis revealed strong seasonal variations in ammonia, with larger contributions from vehicles in the winter months. All ammonia concentrations reported were <3.0 $\mu g/m^3$.

Phan et al. [23] measured ammonia at two urban sites in Seoul, Republic of Korea. They reported hourly data over the year September 2010 to August 2011 acquired using Wavelength Scanned–Cavity Ring Down Spectroscopy (WS-CRDS) ammonia analyzers from Picarro Inc.,Santa Clara, CA, USA. The results show urban ammonia concentrations averaging 10.9 $\pm$ 4.25 ppb and 12.3 $\pm$ 4.23 ppb at the two sites. Diurnal variation in the data highlighted the importance of traffic contributions in the highly congested urban environment from 2010 to 2011. The measurement of atmospheric ammonia concentrations from four sites in Quebec City, Canada, from 2010 to 2013 utilizing passive sampler devices has also been reported [24]. The annual averages at the four sites varied from 0.35 to 17.51 $\mu g/m^3$.

The EPA's first CAFO Final Rule was issued in 2003 and was most recently modified in 2011. Both deal with manure, wastewater and sludge management, but neither address air emissions from CAFOs themselves [25]. As air quality has improved since the 1970s, increasing interest is focused on atmospheric ammonia in both rural and urban sites, as the articles cited above clearly show.

## 2. Materials and Methods

### 2.1. Sampling Sites

The Princess Anne sampling site is on the campus of the UMES Extension Research Farm (38°10′35.67″ N and 75°42′05.17″ W) near Princess Anne in Somerset County (Figures 1 and 2). The site is surrounded by mixed agricultural/wooded and residential acreage, with no chicken houses within a one-mile radius and 7 chicken houses within a two-mile radius. The second site, near Pocomoke City (38°00′50.11″ N, 75°32′43.85″ W), Worcester County, is a mixed agriculture/wooded/industrial/commercial site with a

substantial presence of chicken houses and approximately 1.6 million chickens within a two-mile radius (Figures 1 and 2). In this paper, additional comparisons will be drawn to data from sampling sites MDE operates or has operated at the Horn Point Laboratory (N 38°35′15.09″, W 76°8′27.621″) on the shore of the Choptank River in Dorchester County and to Old Town in Baltimore City (N 39°17′51.839″, W 76°36′16.57″) (Figure 1). The Horn Point site typically has the cleanest air in Maryland and no chickens within a two-mile radius. The Old Town site is urban, with emissions sources typically found in a major metropolitan area. All the sampling sites conform to siting criteria established by the EPA with respect to unrestricted air flow around the site.

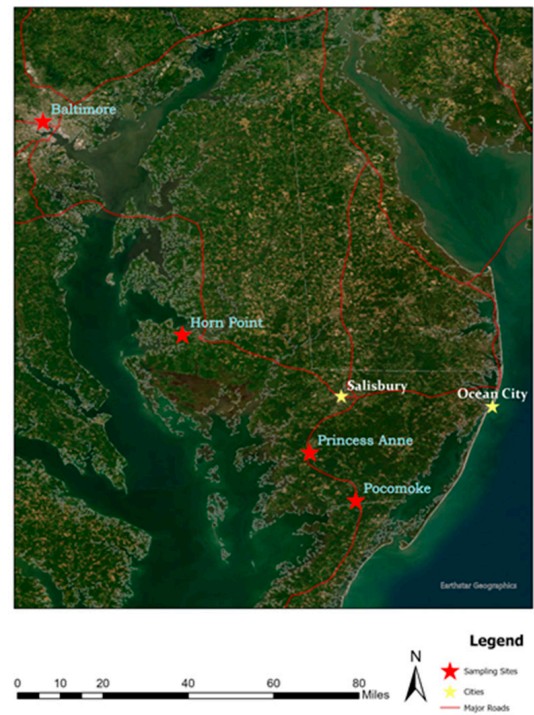

**Figure 1.** Map of sampling sites cited here. From the north: Baltimore Old Town, Horn Point, the Princess Anne site is south of Salisbury, and the Pocomoke site. Old Town and Baltimore County Near Road are collocated on this scale.

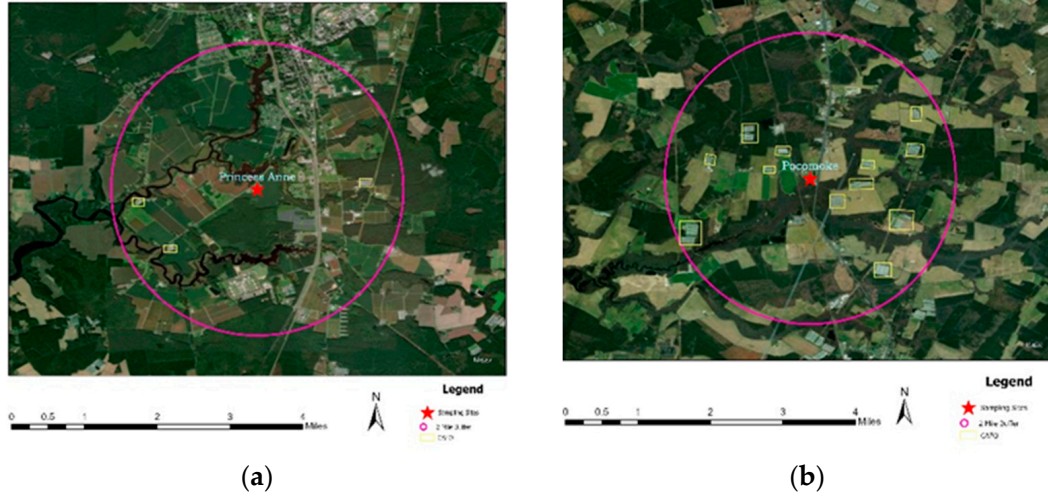

(a)                                                                                                    (b)

**Figure 2.** Two sampling sites indicated by red stars. (**a**) Princess Anne sampling site. (**b**) Pocomoke City sampling site. In each, the pink circle indicates a 2-mile radius. A yellow rectangle indicates a CAFO.

*2.2. Atmospheric Measurements*

Ammonia concentrations were measured in real time using two different EPA-approved techniques at both the Princess Anne and Pocomoke sites. A flowchart showing the methodology for the project is shown in Appendix A, Table A1. A Teledyne-API Model T201 ammonia analyzer (Teledyne Model T201; Teledyne API; San Diego, CA, USA) was used, which measures ammonia rapidly and reliably, even at low concentrations, based on the Federal Equivalent Method (FEM) for nitrogen dioxide. It records concentrations on a minute-by-minute basis, with an expected uncertainty of $\pm 15\%$. The T201 also provides measurements of NO, $NO_2$, NOx and total nitrogen. The Teledyne T201 instrument undergoes periodic precision checks and calibrations in addition to regular preventive maintenance. The equipment is housed in temperature-controlled shelters, and data acquisition can be monitored remotely by MDE and UMES staff to support rapid repairs.

Separate ammonia measurements are made within 10 m of the sampling shelters using passive AMoN samplers, under the auspices of the National Atmospheric Deposition Program (NADP). These Passive Radiello Diffusive Samplers (Catalog Number RAD 168; Sigma-Aldrich, Inc.; St. Louis, MO, USA) are switched out every 2 weeks and are sent to the NADP lab in Wisconsin for independent analysis using a Flow Injection Analysis (FIA) technique. The AMoNs are periodically shipped with travel blanks and include duplicate samplers on a random schedule approximately 4 times per year. The duplicate measurements made at the LES sites over the 33 months reported here reveal an uncertainty of $\pm 6.8\%$. The travel blanks gave averages of 0.08 $\mu g/m^3$ in Princess Anne and 0.09 $\mu g/m^3$ in Pocomoke. The standard deviations for each of the AMoN blanks was 0.03 $\mu g/m^3$. Blanks were not subtracted from the results reported here. The NADP lab posts data on a publicly accessible web site as their analyses are completed [6].

Weather conditions are monitored contemporaneously with the air chemical measurements using Vaisala WXT536 sensors. Temperature, wind direction and wind speed, barometric pressure, relative humidity and rain amounts are reported in hourly increments.

*2.3. Data Reporting*

All data collected in this project are posted on the Lower Eastern Shore Ambient Air Quality Monitoring Project web site, a public website hosted by the Maryland Department of the Environment [1]. A moving twelve-hour average $NH_3$ concentration from each site is also shown on a map of the region using a green–yellow–orange–red–purple–maroon coloration, showing the air quality data often cited in weather reports. At the end of each month, the hour-by-hour averaged data are posted to the MDE web site, accessible to all and downloadable in a variety of formats. Data shown in this paper are from 1 April 2020 to 31 December 2022.

*2.4. Statistical Analysis*

Statistical analyses of all hourly data were performed using SigmaPlot (14.5) software. Data represents the mean $\pm$ standard error (S.E). A Kruskal–Wallis One-Way Analysis of Variance on ranks was used to compare differences in ammonia concentrations from the two sites. Significant differences ($p < 0.05$) were followed by the Dunn's tests. A $p$-value $\leq 0.05$ was considered statistically significant.

Statistical analysis of averaged daily data was performed in Excel (Version 1808) using ANOVA single factor analysis. The Student $t$-test was used to determine whether the means of two sets of measurements were significantly different, or not. Again, a $p$-value $\leq 0.05$ was considered statistically significant.

## 3. Results

*3.1. Atmospheric Data*

3.1.1. Hourly Ammonia Data from Four Sites in Maryland

Hourly ammonia data collected between 1 April 2020 and 31 December 2022 in Princess Anne and Pocomoke are shown in Figure 3. The ammonia levels in Pocomoke are frequently

higher than those in Princess Anne, as is not unexpected from the high density of chicken houses around the site. There were two instances in this time window, however, where the Princess Anne $NH_3$ measurements spiked significantly above the averages for either site. Measurements above 100 ppm were recorded in Princess Anne at 6 AM on 2 July 2020 and from 9 PM on 14 June 2022 through to 8 AM on 15 June 2022. Both spikes in ammonia occurred within days of the application of urea ammonium nitrate (UAN) fertilizer on the University's farm fields immediately adjacent to the sampling station, as confirmed by the farm manager [26]. These fields are planted with corn on even years and with soybeans on odd years. The transient effect of UAN is comparable to, or exceeds, the highest peaks observed in Pocomoke. The farm field immediately adjacent to the sampling site in Pocomoke has been a "no till" field and has been planted with soybeans each year from 2020 to 2022. Plantings in other farm fields within two miles of the sampling sites were not recorded.

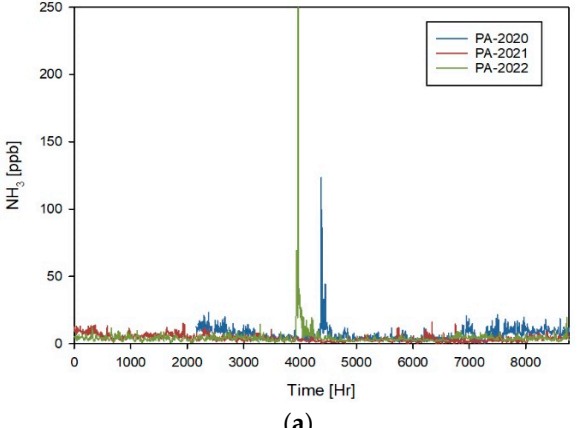

(**a**)

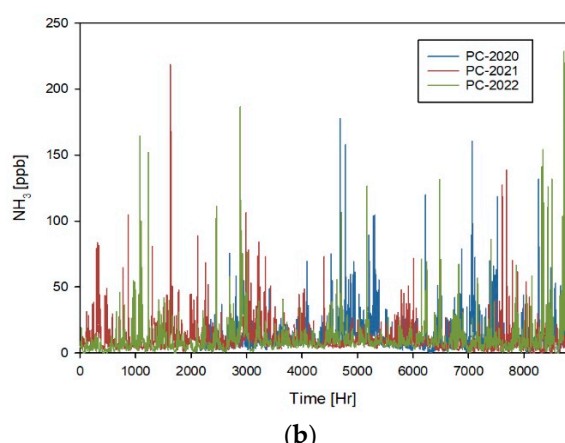

(**b**)

**Figure 3.** Ammonia concentration by hour in (**a**) Princess Anne (PA), an area of low chicken house density, 1 April 2020 to 31 December 2022. The maximum value on 15 June 2022 at 6 am was actually 336.2 ppb; all other measurements were below 250 ppb, as shown. (**b**) Pocomoke (PC), an area of high chicken house density, 1 April 2020 to 31 December 2022.

For comparison, $NH_3$ measurements in Maryland from Baltimore Old Town and Baltimore County Near Road (both urban, high-traffic areas west of the Chesapeake Bay) and Horn Point (a rural, non-farm, waterfront location on the Eastern Shore) are shown in Figure 4a,b. Note the change in vertical scales, which emphasizes that ammonia concentrations in these non-agricultural areas are significantly lower than those measured in LES agricultural areas.

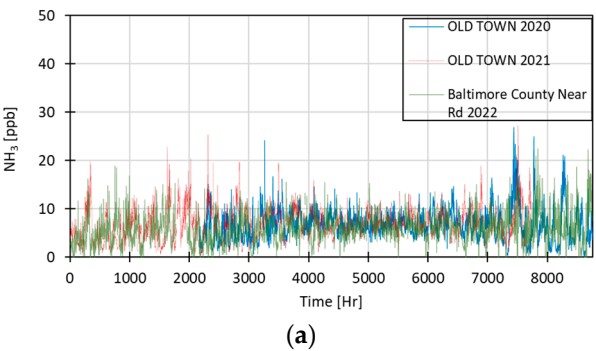

(**a**)

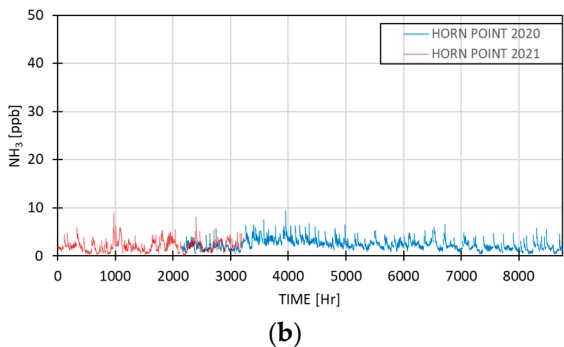

(**b**)

**Figure 4.** Ammonia concentration (ppb) by hour in (**a**) Old Town and from Baltimore County Near Road, both in urban, high-traffic areas, 1 April 2020 to 31 December 2022, and (**b**) in Horn Point, a rural area without chicken farms, 1 April 2020 to 14 May 2021. The MDE discontinued ammonia measurements at Horn Point on 14 May 2021 following years of consistently low readings. Note change in vertical scale in comparison to Figure 3a,b.

### 3.1.2. Bi-Weekly AMoN Data

In addition to the hourly measurements, passive AMoN samplers were used to measure $NH_3$ concentrations on a two-week average basis in both Princess Anne and Pocomoke. The biweekly data from the AMoN samplers as compared to the two-week averages calculated from the hourly measurements by the T201 instruments are shown for Princess Anne in Figure 5a and for Pocomoke in Figure 5b.

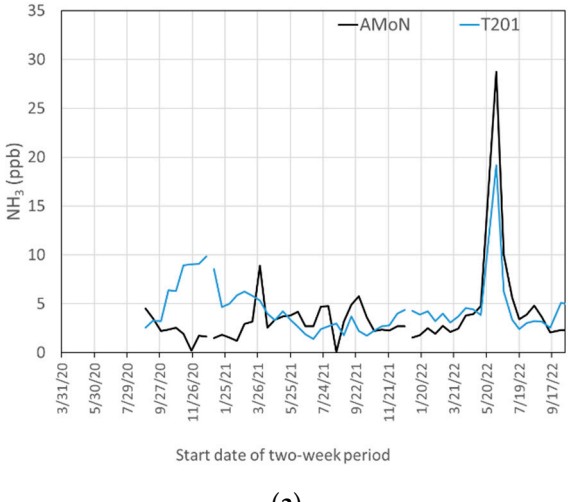

(**a**)

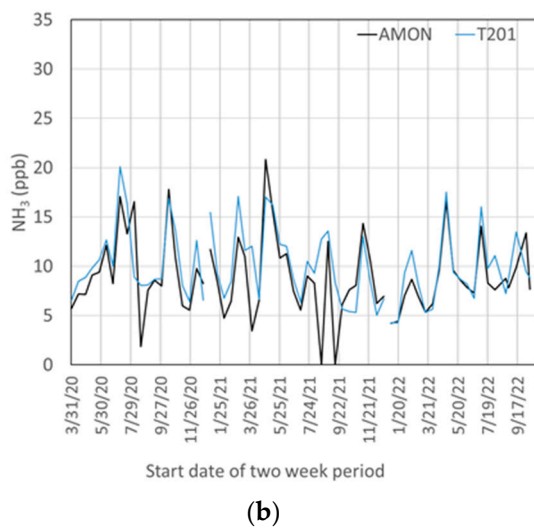

(**b**)

**Figure 5.** AMoN two-week average $NH_3$ concentrations vs. T201 two-week average $NH_3$ concentrations at the (**a**) Princess Anne site (the peak on 5 June 2022 reflects the application of UAN fertilizer, as discussed) and the (**b**) Pocomoke site. The AMoN data are converted from $\mu g/m^3$ to ppb using a conversion factor of 0.695 $(\mu g/m^3)/ppb\ NH_3$.

### 3.2. *Annual Analysis of Ammonia Data from Princess Anne and Pocomoke*
#### 3.2.1. Ammonia Analysis

The annual mean data sets generated from examining the hourly measurements of ammonia over the first 33 months are presented here (Table 1). Figure 6 illustrates that the ammonia concentrations varied between the two LES sites and across the three years in which data were collected. Ammonia concentrations were significantly different in each year from 2020 to 2022 at both the Princess Anne and Pocomoke sites ($p < 0.001$; Figure 6). Additionally, concentrations of ammonia were significantly higher in PC than in PA for each of the three years.

**Table 1.** Summary statistics for annual $NH_3$ measured from the sampling sites from 1 April 2020 to 31 December 2022.

| Site | Year | Mean | Std. Dev | Std. Error |
|------|------|------|----------|------------|
| Princess Anne | 2020 | 6.374 | 5.049 | 0.0649 |
| Pocomoke | 2020 | 11.347 | 11.457 | 0.147 |
| Old Town | 2020 | 6.700 | 3.100 | 0.0401 |
| Horn Point | 2020 | 2.300 | 1.000 | 0.0129 |
| Princess Anne | 2021 | 3.744 | 2.296 | 0.0255 |
| Pocomoke | 2021 | 10.262 | 10.841 | 0.121 |
| Old Town | 2021 * | 6.800 | 3.100 | 0.0364 |
| Horn Point | 2021 * | 1.900 | 1.100 | 0.0204 |
| Princess Anne | 2022 | 4.388 | 8.810 | 0.0972 |
| Pocomoke | 2022 | 9.790 | 11.759 | 0.130 |

* Limited data sets (Old Town covered the period from 1 January 2021 to 19 November 2021; Horn Point covered the period from 1 January 2021 to 14 May 2021).

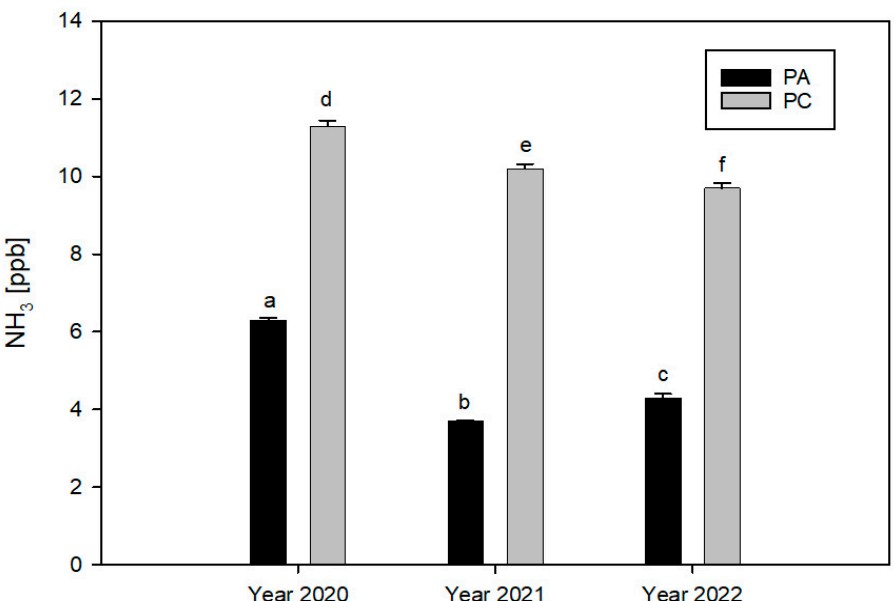

**Figure 6.** Mean ± standard error (S.E.) $NH_3$ levels (ppb) by year in Princess Anne (PA) and Pocomoke (PC). The letter designations indicate that each data set is statistically significantly different from each of the others.

### 3.2.2. AMoN Results Compared to T201 Results

The results of the AMoN measurements were consistent with the two-week average values from the Teledyne T201. Once the T201 in Princess Anne was passivated, and for the entire length of the project in Pocomoke, the AMoN measurements are 83.3% of the two-week averaged T201 measurements. The differences by year are shown in Table 2.

**Table 2.** Results of average difference calculations. Differences between T201 and AMoN two-week average values ($\delta$ = T201-AMoN) calculated in ppb and compared to the average AMoN readings for each time period in Princess Anne (PA) and Pocomoke (PC). Positive values indicate T201 average > AMoN average for that time period.

| Data | Start Dates | PC AMoN Average (ppb) | PC Avg $\delta$ (ppb) | PA AMoN Average (ppb) | PA Avg $\delta$ (ppb) |
|---|---|---|---|---|---|
| all 2020 * | 31 March 2020 to 22 December 2020 | 9.51 | 1.00 | 2.30 | 4.21 |
| all 2021 | 5 January 2021 to 21 December 2021 | 8.71 | 2.22 | 3.20 | 0.49 |
| all 2022 | 4 January 2022 to 12 October 2022 | 8.56 | 0.66 | 4.72 | −0.09 |
| project so far | 31 March 2020 to 12 October 2022 | 8.90 | 1.07 | 3.60 | 0.89 |

* AMoN data for PC covered the period from 31 March 2020 to 22 December 2020 and PA from 1 September 2020 to 22 December 2020 in the year 2020.

### 3.3. Impact of the COVID-19 Pandemic on Data

The earliest data collected in this project were reviewed to determine whether data were affected by the statewide mandatory shut down of non-essential businesses implemented in Maryland on 20 March 2020 in response to the COVID-19 epidemic [27]. The mandatory shut down was lifted on 15 May 2020, and non-essential businesses were permitted to reopen [28]. Agriculture was considered an essential business and permitted to continue operations during the shutdown. The means of the hourly ammonia measurements made in Princess Anne and Pocomoke from 1 April, the first day of data acquisition in 2020, through 15 May 2020 are shown in Table 3. Also shown are the means of data

collected at both sites over the same time window, from 1 April to 15 May, post-pandemic in 2021 and 2022. The letters indicate whether the data are statistically the same (same letters) or different (different letters) as those collected in other years.

**Table 3.** Mean ± standard error of hourly ammonia concentrations (ppb) acquired in Princess Anne and Pocomoke during the mandatory shutdown of non-essential business from 1 April 2020 to 15 May 2020 and the same dates in 2021 and 2022. The letters indicate whether the data between the two sites and among the three years are statistically the same (same letter) or different (different letters).

|  | 2020 | 2021 | 2022 |
|---|---|---|---|
| Princess Anne | 8.9 ± 0.1 (a) | 3.9 ± 0.06 (b) | 4.1 ± 0.04 (b) |
| Pocomoke | 6.5 ± 0.2 (c) | 9.2 ± 0.3 (a, d) | 7.3 ± 0.4 (c) |

## 4. Discussion

The primary objective of this project was to determine the ambient ammonia concentrations at two sites on the Lower Eastern Shore of Maryland. The sites are comparable, insofar as possible, with respect to all factors except for one: the density of broiler chicken CAFOs within a two-mile radius. Overall, the ammonia levels in Pocomoke, Worcester County, where CAFO density is high, were found to be significantly higher than those in Princess Anne, Somerset County. The exception was during short time intervals following the application of UAN fertilizer on the fields adjacent to the PA sampling site, as shown in Figure 3a. Ammonia levels at Old Town and Horn Point are substantially lower than those in either Princess Anne or Pocomoke.

In Princess Anne and Pocomoke, the two-week average $NH_3$ concentrations measured with the Teledyne T201 device were expected to be quite similar to those obtained from the two-week average measurements using the AMoN samplers. This was observed once the initial passivation of the T201 device was accomplished. Passivation occurred over the time window from 12 October 2020 through 2 March 2021 in Princess Anne. Passivation is known to be required and is considered responsible for the early discrepancies between the Teledyne and the AMoN measurements in Princess Anne. The Pocomoke site came on-line before the Princess Anne site, and all values reported in this paper were recorded after the Pocomoke instrument was passivated. For all the measurements reported in Pocomoke, and for those in Princess Anne after 2 March 2021, measurements made by the AMoN samplers made up 83.3% of those reported by the T201. In Figure 5a,b, when duplicate AMoN samplers were deployed for the same two-week period, the average value is used in preparing the plots. The largest difference between simultaneous AMoN measurements was 6.8%. This difference was recorded in Princess Anne for the two-week window beginning 7 June 2022. Recall that the hourly data showed a spike in $NH_3$ concentration due to UAN fertilizer applications that began on 15 June 2022, as shown in Figure 3a. AMoN measurements had not yet started at the Princess Anne site during the ammonia peak in 2020. In Pocomoke, the AMoN results consistently align with those from the T201 hourly measurements. Both techniques show that Pocomoke in Worcester County has higher $NH_3$ concentrations than Princess Anne in Somerset County.

In 2021 and 2022, $NH_3$ levels in PA were similar to levels reported in other rural agricultural areas in the United States. Saylor et al. reported a 6-month mean value from 5 min resolution gas-phase ammonia concentration measurements of 3.2 ± 2.37 ppbv in rural Yorkville, GA, from July to December 2007 [29]. Yorkville is described as a mixed agricultural and forested area, with broiler houses located distances of 1.3 to 3 km to the southeast of the sampling site. Wind sector data suggested that the $NH_3$ values from Yorkville were largely influenced by the poultry houses. Ammonia levels in year 2020 from PA were comparable to those reported by Zbieranowski and Aherne from southern Ontario, Canada [30]. The authors reported mean ammonia values of 3.56 µg/m$^3$ for 18 sites described as mixed residential/agricultural occupying a 15 km × 15 km sector.

Measurements were made from 30 March 2010 to 29 March 2011 using Willems badge passive samplers.

The levels of ammonia in PC measured across the 33 months reported here were also consistent with those observed in other studies. Concentrations of $NH_3$ in rural communities near Bejing, China, measured using Ogawa passive samplers were reported to be $4.5 \pm 2.6$ ppb (2007), $6.6 \pm 7.0$ ppb (2008), $7.1 \pm 3.5$ ppb (2009) and $14.2 \pm 10.8$ ppb (2010) by Meng et al. [31]. The higher mean in 2010 was attributed to the effects of high temperatures. The site was described as surrounded by mixed farmland, orchards and forests in a region of rolling hills, lacking CAFOs. Wang et al. used the monitoring instrument for AeRosols and GAses (MARGA) to measure rural ammonia at Diashan Lake near Shanghai, China, and compared their results to nearby urban and industrial sites [32]. The annual mean rural ammonia concentration from 1 July 2013 to 30 June 2014 (excluding January and February 2014) was $12.4 \pm 9.1$ ppb, with a peak of 79.4 ppb. These results were higher than their urban results, $6.2 \pm 4.6$ ppb, but lower than the industrial mean of $17.6 \pm 9$ ppb they reported. Rice was indicated as the most important crop in the rural environment studied. Overall $NH_3$ levels at both LES sites were lower than in rural sites in the North China Plain, as reported by Shen et al. in 2011 [33].

Evidence of the COVID-19 pandemic shutdown can be seen in some of the data here. The state of Maryland implemented a shutdown of non-essential businesses from 20 March 2020 through 15 May 2020 because of the COVID-19 pandemic [27,28]. Farming was excluded from the shutdown as an essential business. Analysis of the hourly $NH_3$ levels over the COVID-19 shutdown time window showed that the average ammonia levels in Pocomoke were significantly lower than those in Princess Anne in 2020, but not in 2021 or 2022, as shown in Table 3. Atmospheric ammonia concentrations have been observed to be correlated with temperature [34], so the average daily temperatures from 1 April to 15 May 2020 were compared with the average daily temperatures from 1 April to 15 May 2021 and to the average daily temperatures from 1 April to 15 May 2022 for both sites to ensure that temperature was not an important variable in determining these results. Temperatures from 1 April through 15 May in Princess Anne were not statistically different from one another in 2020, 2021 and 2022. The means of the average daily temperatures in Princess Anne were $54.1 \pm 1.0$ °F in 2020, $56.5 \pm 1.2$ °F in 2021 and $55.9 \pm 1.1$ °F in 2022. The means of the average daily temperatures in Pocomoke were $53.5 \pm 1.0$ °F in 2020, $56.0 \pm 1.2$ °F in 2021 and $55.5 \pm 1.0$ °F in 2022. None of these temperatures are statistically significantly different from any of the others. Therefore, temperature cannot be considered to have affected the ammonia concentrations at PA or PC during the COVID-19 shutdown, as compared to the same time window in 2021 or 2022.

The DCA confirms that there was not a significant reduction in the production of chicken in southwest Worcester County during the period of the stay-at-home order, although there may have been some longer layouts (the time lag until a grower obtains a new set of birds) due to logistical adjustments [35]. The effects of the COVID-19 shut down and the reduction in daily traffic on a variety of ambient air components have been reported. Cui reported the impact of the COVID-19 shutdown on ammonia across India using the XGBoost model and data from the Central Pollution Control Board and IASI, including meteorological data from the European Centre for Medium Range Weather Forecast reanalysis [36]. Cui found that ammonia emission decreases from the industrial and traffic sectors associated with the shutdown were offset by increases from agricultural emissions due to higher temperatures over that shutdown time window in Lucknow. Similar ammonia emission reductions in other cities were offset by increases in residential emissions. A study of ammonia concentrations during the winters of 2019 and 2020 in Beijing, China, by Zhang et al. [37] reported the impact of the Chinese New Year Holiday and the COVID-19 shutdown and commented that the 2020 data were significantly higher compared to historical data from the same sites in 2017. Their measurements were made at five sites around Beijing, four being along major roadways, which were expected to be directly affected by traffic emissions, and one in a residential/campus area, which was

not expected to be affected by traffic flow. Weekly measurements using ALPHA passive samplers allowed the determination of the N-isotopic distribution in $NH_3$. The STAR model was then used to determine the contributions of specific sources to the $NH_x$ concentrations in the air at the five sites. Although vehicle traffic decreased significantly during the pandemic, significant reductions in $NH_3$ were not reported from any site. They concluded that meteorological factors over-rode the reduction in observed vehicle emissions.

Traffic volumes may have impacted the Pocomoke data reported here. The Pocomoke sampling site is located only ~150 m from Maryland State Route 13, a major north–south thoroughfare on the Eastern Shore. There is a clear sight line from the sampling station to the highway. Though there is no expectation that farming activities were reduced during the shutdown, there is information that traffic passing the sampling site was reduced by the shutdown. Traffic data from the Maryland Department of Transportation (MDoT) indicate that annual average daily traffic monitored at ATR#37, on Route 13 north of the Virginia State line, declined 15% in 2020 compared to 2019 from 20,686 to 17,523 [38]. Traffic at this site in 2021 recovered to within 1.5% of its value in 2019. The 2022 data are not yet posted. The sampling location is less than 1 mile north of the state line, with two country roads, but no major thoroughfares, intersecting Rt 13 between the PC sampling site and the Virginia state line. In contrast, traffic volumes on Stewart Neck Road, adjacent to the Princess Anne location, are expected to be significantly lower than on Rt 13, as it is a local road, not connected to any other major thoroughfares. Traffic on Stewart Neck Road is not measured by the MDoT. The lower concentration of ammonia in Pocomoke during the pandemic is therefore consistent with the reduction in traffic volume reported on Rt 13, but the reductions in pollutants cannot be unambiguously attributed to lower traffic volumes. There is no evidence that the pandemic stay-at-home order had any discernable effect on ammonia levels in Princess Anne. The COVID-19 shutdown highlights the variety of factors which affect ambient air quality, even in the vicinity of large numbers of CAFOs.

The sites of the sampling stations we have utilized in Somerset County and Worcester County may be contrasted from an Environmental Justice (EJ) perspective. As illustrated in Table 4 and Figure 7, Somerset County has the highest percentage of an African American population on the shore. It also has the lowest average household income and the highest percentage of persons living in poverty in the state of Maryland for the ten years covered by the 2020 census [39].

**Table 4.** Some census characteristics of Worcester and Somerset Counties, Maryland. The Pocomoke City site is in western Worcester County. The Princess Anne sampling site is in Somerset County [39].

|  | Worcester County | Somerset County |
|---|---|---|
| Population, Census, 1 April 2020 | 52,460 | 24,620 |
| Population, Census, 1 April 2010 | 51,454 | 26,470 |
| % Persons under 18 years | 17.1% | 17.1% |
| % Persons 65 years and over | 28.2% | 17.4% |
| % Households with a computer, 2016–2020 | 90.8% | 86.2% |
| Median household income (in 2020 USD), 2016–2020 | $65,396 | $44,980 |
| % Persons in poverty | 11.7% | 22.2% |

At first glance, exposure to high levels of ambient ammonia does not appear correlated with either non-white population rates or with poverty rates on the shore. However, as shown in Table 5, Worcester County is a diverse county. Geographically, it extends from the Delaware border to the Virginia border along the Atlantic Ocean, including all of Maryland's coastal bays. Ocean City, in the northeast, is a major tourist destination. In Worcester County, Census track 9510, Berlin is a residential area with a lively downtown which draws significant tourist traffic. The area around Pocomoke is primarily agricultural, with some industrial, some commercial and some green space along the main thoroughfare, Route 13. The overall Worcester County demographics are influenced by the older, wealthier, high density population areas to the north and east. The area around the sampling site is

described by Census Track 9515, which is not characterized well by county-wide data. The variety of local characteristics in Worcester County highlights the importance of identifying an appropriate sub-county Census Track to identify the characteristics of residents most exposed to ambient levels of $NH_3$.

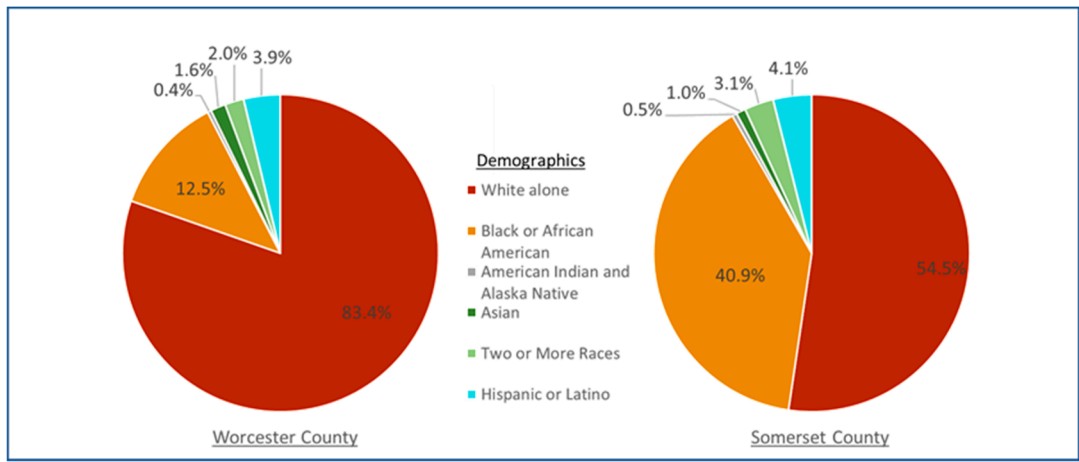

**Figure 7.** Ethnic demographics of Worcester and Somerset Counties as of the 2020 census [39].

**Table 5.** Census Track-specific data for sites including the sampling stations in Pocomoke, in contrast to selected surrounding Census Tracks in Worcester County [40].

|  | **Pocomoke Site** | **Berlin** | **Ocean City** |
|---|---|---|---|
| Geographic Area Name | Census Tract 9515, Worcester County, Maryland | Census Tract 9510, Worcester County, Maryland | Census Tract 9500, Worcester County, Maryland |
| Census Tract Identifier | 24047951500 | 24047951000 | 24047950000 |
| Percent Minority | 47.1 | 29.2 | 7.9 |
| Percent Poverty | 45.3 | 35.6 | 33.23 |
| Percent Limited English Proficiency | 1.2 | 0.6 | 1.1 |
| Socioeconomic Score Percent (For this Tract) | 31.2 | 21.8 | 14.08 |
| Socioeconomic Score % (Distribution across Maryland) | 67.89 | 51.69 | 33.33 |

The Pocomoke sampling site is located in Census Track 9515. The difference in socioeconomic characteristics of residents of that census track from those in the north and eastern part of Worcester County is clearly indicated by the characteristics shown in Table 5. Table 6 shows that the Princess Anne site, by virtue of its position west of Route 13, has a comparable poverty rate to the Pocomoke site. The percentage of residents identifying as minority ethnicities is significantly higher around the Pocomoke site, which also has a higher socioeconomic score. A serious evaluation of the EJ impact on the analysis of our data is outside the bounds of our expertise. The data here are provided to highlight the complications of identifying sampling sites for other scientists interested in pursuing answers to the challenging questions that EJ poses.

**Table 6.** Census Track-specific data for sites including the sampling stations in Princess Anne, in contrast to selected surrounding Census Tracks in Somerset County [40].

|  | **E of Princess Anne** | **Princess Anne Site** | **N of Pocomoke City** | **W of Site to Bay** |
|---|---|---|---|---|
| Geographic Area Name | Census Tract 9301.01, Somerset County, Maryland | Census Tract 9301.02, Somerset County, Maryland | Census Tract 9303, Somerset County, Maryland | Census Tract 9302, Somerset County, Maryland |
| Census Tract Identifier | 24039930101 | 24039930102 | 24039930300 | 24039930200 |
| Percent Minority | 75.8 | 31.5 | 26.2 | 18.4 |
| Percent Poverty | 60.62 | 45.76 | 38.34 | 36.54 |
| Percent Limited English Proficiency | 1.8 | 1.9 | 0 | 0.7 |
| Socioeconomic Score Percent (For this Tract) | 46.07 | 26.39 | 21.51 | 18.55 |
| Socioeconomic Score % (Distribution across Maryland) | 91.65 | 59.68 | 51.04 | 43.99 |

## 5. Conclusions

The results reported here provide measurements of ambient air quality on the Lower Eastern Shore, which, by virtue of its location between the Chesapeake Bay and coastal bays that empty into the Atlantic Ocean, is a region of significant environmental importance. The CAFOs on the Lower Eastern Shore primarily house broiler chickens. The number of CAFOs on the shore has varied over the past 25 years. However, the live weight of chickens produced on Delmarva has increased almost continuously, approximately doubling from 1987 to 2018, when it reached 4.3 billion pounds [41]. We anticipate that the data reported here will provide an improved picture of the contributions of poultry CAFOs to ambient ammonia levels on the LES. Although the annual ammonia levels in Pocomoke are statistically significantly higher than those in Princess Anne, there is no indication that ammonia concentrations in Pocomoke or Princess Anne pose any hazard to human health based on any current standards.

Further evaluation of this data set is in process, and an extension and expansion of the project through 2024 is underway. Extended data acquisition and expanded data analysis are expected to reveal improved information about the influence of farm crop rotation on ammonia levels. Further analysis will provide information about the influence of weather factors on ambient levels of ammonia in this distinctive coastal environment, as well as permit investigation of the correlations, if any, between ammonia and particulate matter concentrations. One limitation of this work is that no anion concentrations are being measured; another is that the chemical compositions of the atmospheric at the two sampling sites are being considered to represent extremes in ammonia concentrations to which the public is typically exposed on the LES due to the presence of CAFOs.

In summary, the presence of high numbers of poultry CAFOs on the Eastern Shore of Maryland does contribute to higher ambient levels of atmospheric ammonia. Since ammonia is not a criteria pollutant, no nation-wide environmental standards for it have yet been established for ambient air. The Maryland Department of the Environment has established 350 ppb/hour and 250 ppb/8 h average as its action levels for ammonia. These limits were established in consideration of a wide variety of health and workplace recommendations and allowing for a safety factor of 100. Over the first 33 months of this project, none of the hourly observations showed readings in excess of 350 ppb, nor did the observations produce any 8 h average readings in excess of 250 ppb.

**Author Contributions:** B.B. and D.G.S. contributed equally to this project. Data curation, B.B.; formal analysis, B.B. and D.G.S.; funding acquisition, D.G.S.; investigation, B.B. and D.G.S.; resources, B.B. and D.G.S.; validation, B.B.; writing—original draft, D.G.S.; writing—review and editing, B.B. and D.G.S. All authors have read and agreed to the published version of the manuscript.

**Funding:** This research was funded by the Delmarva Chicken Association (DCA, formally DPI) and The Keith Campbell Foundation for the Environment. "The funders had no role in the design of the project; in the collection, analyses, or interpretation of data; in the writing of the manuscript; or in the decision to publish the results".

**Data Availability Statement:** All data collected in this project are posted and available to the public. See: Maryland Department of the Environment Lower Eastern Shore Ambient Air Quality Monitoring Project https://mde.maryland.gov/programs/air/AirQualityMonitoring/Pages/Lower-Eastern-Shore-Monitoring-Project.aspx (accessed on 18 July 2023). And: National Atmospheric Deposition Program Ammonia Monitoring Network (AMoN) Sites MD92 and MD97, https://nadp.slh.wisc.edu/networks/ammonia-monitoring-network/ (accessed on 18 July 2023).

**Acknowledgments:** The authors gratefully acknowledge contributions from Moses Kairo, who supported our participation in this project; Ryan Auvil, Air Monitoring Program Air and Radiation Administration, Maryland Department of the Environment; Alex Echols, Campbell Foundation; Holly Porter, DCA; Ali Ishaque for access to SigmaPlot 14.5 and Chelsea Richardson, for technical consultation about Sigma Plot; and Tracie Bishop for GIS support.

**Conflicts of Interest:** The authors declare no conflict of interest.

# Appendix A

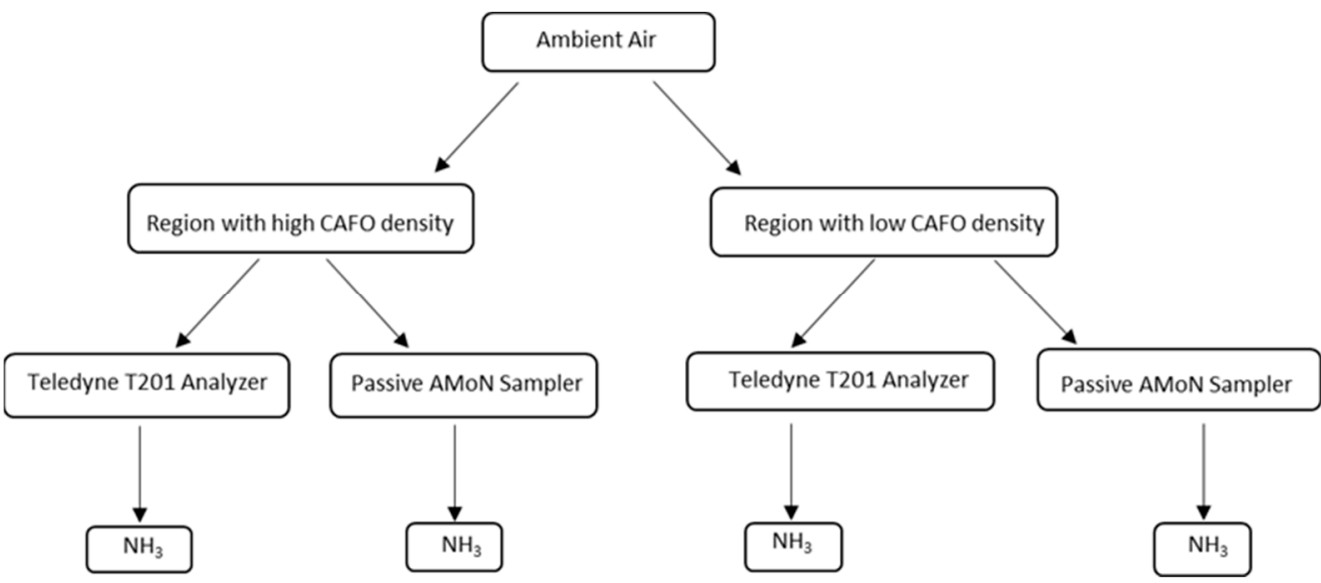

**Figure A1.** Flowchart showing the methodology for the project.

**Table A1.** List of Abbreviations.

| | |
|---|---|
| ACGIH | American Conference of Governmental Industrial Hygienists |
| AERMOD | American Meteorological Society (AMS)/U.S. EPA Regulatory Model |
| ALPHA | CEH Adapted Low-Cost Passive High Absorption |
| AMoN | Ammonia Monitoring Network |
| CAFO | Concentrated Animal Feeding Operation |
| CDC | Centers for Disease Control |
| DCA | Delmarva Chicken Association |
| EJ | Environmental Justice |
| EPA | Environmental Protection Energy |
| LES | Lower Eastern Shore of Maryland- Wicomico, Somerset and Worchester counties |
| MARGA | Monitoring instrument for AeRosols and Gases |
| MDE | Maryland Department of the Environment |
| MDoT | Maryland Department of Transportation |
| NAAQS | National Ambient Air Quality Standards |
| NAEMS | National Air Emissions Monitoring Study |
| NADP | National Atmospheric Deposition Program |
| $NH_3$ | Ammonia |
| NIOSH | National Institute for Occupational Safety and Health |
| NOAEL | No Observed Adverse Level |
| NRC | National Research Council |
| OSHA | Occupational Health and Safety Administration |
| Ppb | parts-per-billion |
| Ppbv | parts-per-billion by volume |
| Ppm | parts-per-million |
| REL | Recommended Exposure Limit |
| SEARCH | Southeastern Aerosol Research and Characterization study |
| STEL | Short-Term Exposure Limit |
| TLV | Threshold Limit Value |
| UAN | Urea ammonium nitrate (fertilizer) |
| UMES | University of Maryland Eastern Shore |

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
