# Peer review of "First Measurement of Ambient Air Quality on the Rural Lower Eastern Shore of Maryland"

_agronomy, doi:10.3390/agronomy13071952_

Round 1

Reviewer 1 Report

Comments:

1. Add more numerical results in Abstract.

2.      Add a scale to Figure 1.

3.      What is 月in Figure 20? pl. check.

4.      Title for a table throughout the manuscript should be above each table.

5.      Compare your results with those in literature.

6.      Add limitations of your study.

7.     To improve the paper regrading PMs, the authors can read and use the following papers:

        I.            Exposure to high levels of PM2. 5 and PM10 in the metropolis of Tehran and the associated health risks during 2016–2017

     II.            Levels of particulate matters in air of the Gonabad city, Iran

  III.            Particulate matters and bioaerosols during Middle East dust storms events in Ilam, Iran

Author Response

Comments from Reviewer 1

Comment 1: Add more numerical results in Abstract

Response: Thank you. We agree with this comment and have added the numerical results in the Abstract. This change can be found on page 1, first paragraph, line 24.

Comment 2: Add a scale to Figure 1.

Response: Agree. We have added a scale to Figure 1. This change can be found on page 4, line 138.

Comment 3: What is in Figure 20? Pl. check.

Response: Thank you. We have removed Figure 20 from the manuscript so as to reduce the number of figures in this manuscript.

Comment 4: Title for a table throughout the manuscript should be above each table

Response: Agree. We have revised accordingly. The changes can be found on page 8, line 257; page 9, line 269; page 10, line 284; page 12, line 388; page 14, lines 425 and 429.

Comment 5: Compare your results with those in literature.

Response: Agree. We have modified the discussion to include your suggestion. The changes can be found on pages 11 and 12, lines 314– 333.

Comment 6: Add limitations of your study

Response:  Thank you. We agree with this comment and have added the limitations of our study. This change can be found on page 15, paragraph 2, lines 441-444.

Comment 7: To improve the paper regarding PMs, the authors can read and use the following papers.

  1. Exposure to high levels of PM2.5 and PM10 in the metropolis of Tehran and the associated health risks during 2016-2017.
  2. Levels of particulate matters in air of the Gonabad city, Iran
  • Particulate matters and bioaerosols during Middle East dust storms events in IIam, Iran.

Response: Thank you for this suggestion. We have removed the PM data from the manuscript so as to reduce the number of figures in this manuscript.

Reviewer 2 Report

The manuscript ‘First Measurement of Ambient Air Quality on the Rural Lower Eastern Shore of Maryland’ was carried out to report the air quality near the agricultural area. The work is very meaningful. However, it looks like a preliminary scientific report, and some drawbacks must be resolved. The results only provided data and comparisons but lacked other statistics and model analyses. There are too many figures in the manuscript, and I suggested the authors could merge figures which include similar information. Similar to the writing, there are too many paragraphs, and I suggested that the authors improve and concentrate on the key points.

English is well written, but the text structure is not good.

Author Response

Comments from Reviewer 2

Comments: The manuscript ‘First Measurement of Ambient Air Quality on the Rural Lower Eastern Shore of Maryland’ was carried out to report the air quality near the agricultural area. The work is very meaningful. However, it looks like a preliminary scientific report, and some drawbacks must be resolved. The results only provided data and comparisons but lacked other statistics and model analyses. There are too many figures in the manuscript, and I suggested the authors could merge figures which include similar information. Similar to the writing, there are too many paragraphs, and I suggested that the authors improve and concentrate on the key points.

Response: We have merged figures with similar information. This change can be found on page 5, line 141 (Figure 2); page 7, lines 204 (Figure 3); page 7, line 211 (Figure 4); page 8, line 228 (Figure 5). We have also included a summary statistics table (Table 1). Thank you for your suggestion on the inclusion of the model analyses. Even though it sounds plausible to have considered this, our primary objective was to determine how CAFOs influence air quality on the LES. We hope our paper will be helpful to modelers in the future. We have improved and concentrated on the key points.

Comments on the Quality of English Language

English is well written, but the text structure is not good.

Response: The document has been thoroughly edited.

Reviewer 3 Report

The study is interesting. However, the whole expression is unclear, and the focus is not prominent. The structure is unreasonable. I cannot recommend it to be published in its current state. The followings provide some comments for further study.

1)      What are the innovation and motivation of this study? Please explain and present them in detail.

2)      The introduction section is too general. It is suggested that the author adequately cite previous studies and highlight the motivation of the study.

3)      Inadequate references were used in the introduction. The author should select appropriate references and highlight the research features according to the content of the study. Please develop the literature.

4)      Due to the problems mentioned above in the introduction, this section should be completely restructured.

5)      The authors need to revise the abstract by including the research problem, objective, main results, and research recommendations that are practically attainable.

6)       Keywords should not overlap with the title.

7)      Please provide detailed information for measurements used in this study. (Descriptive statistics)

8)      I could not see the meaning of some abbreviations in the equations in the article. Please provide a nomenclature.

9)      Please provide a flow chart so that readers can easily understand this study.

10)  Provide detailed information about the equipment used in this study (e.g., accuracy, model, country)

11)  There are too many figures, some of them can be included in the appendix.

12)  Please edit all tables according to journal style.

13)  Limitations and future directions are not included in this form.

14)  The conclusions should be appropriately simplified and highlight the conclusions and insightful findings of the study.

15)  Although the subject of the article is interesting, there are significant problems in conveying its focus, interpreting it, and highlighting its novelty. 

Minor editing of the English language required

Author Response

The study is interesting. However, the whole expression is unclear, and the focus is not prominent. The structure is unreasonable. I cannot recommend it to be published in its current state. The followings provide some comments for further study.

  • What are the innovation and motivation of this study? Please explain and present them in detail.

The motivation for this study is to investigate the variation in ambient levels of atmospheric ammonia between two relatively well characterized environments on the Lower Eastern Shore of Maryland- one in a CAFO intensive region and one in an agricultural region w/out CAFOs. The LES is an environmentally sensitive region between the Atlantic Ocean and the Chesapeake Bay.  Current models lack support of actual experimental measurements in too many cases. (Nair and Yu, Atmosphere 2020(11) 1092.) No measurements have been made of air quality regionally until this project was implemented.

  1. The introduction section is too general. It is suggested that the author adequately cite previous studies and highlight the motivation of the study.

We have added several references to previous studies in the introduction and clarified the motivation for this work.

  • Inadequate references were used in the introduction. The author should select appropriate references and highlight the research features according to the content of the study. Please develop the literature.

We have added several references to previous studies to the introduction and highlighted the content in contrast to this work.

  • Due to the problems mentioned above in the introduction, this section should be completely restructured.

Done.

  • The authors need to revise the abstract by including the research problem, objective, main results, and research recommendations that are practically attainable.

We have revised the abstract to address the reviewer’s concerns.

  • Keywords should not overlap with the title.

Thank you!  We have modified the key words.

  • Please provide detailed information for measurements used in this study. (Descriptive statistics)

We have added the descriptive statistics for the study. See Table 1 in section 3.2.1.

  • I could not see the meaning of some abbreviations in the equations in the article. Please provide a nomenclature.

We have reviewed the acronym list.  We have removed all equations from the paper.  The value delta of the difference between the two-week average T2001 ammonia measurements and the two-week average AMoN measurements. The value of delta, d, is defined in the description for Table 2.

  • Please provide a flow chart so that readers The can easily understand this study.

Added in Appendix A.

  • Provide detailed information about the equipment used in this study (e.g., accuracy, model, country)

We have added more details about the equipment used and the accuracy of each. See page 5, lines 152-171.

There are too many figures, some of them can be included in the appendix.

We have combined similar figures and deleted others.

  • Please edit all tables according to journal style.

Done.

  • Limitations and future directions are not included in this form.

Please see page 15, lines 437-444.

14)  The conclusions should be appropriately simplified and highlight the conclusions and insightful findings of the study.

            We have simplified the conclusions and highlighted the key findings of this project.

15)  Although the subject of the article is interesting, there are significant problems in conveying its focus, interpreting it, and highlighting its novelty. 

            We have endeavored, by addressing each of the specific reviewer comments, to improve the focus, clarify the interpretations of results and highlight the role of this project in providing accurate experimental results which be used to revise emission inventories and improve ammonia modeling efforts in the future.  Further analysis of this data set is underway.

Comments on the Quality of English Language Minor editing of the English language required

               The document has been extensively edited in response to other reviewer comments.

Reviewer 4 Report

The topic of the paper is interesting and the submitted article is well processed. Specifically, the problem under the investigation is specified, relevant literature overview is provided and the objectives are also relevant and rational. The proposed methodology is also rational one. The results part of the paper is interesting and provided discussion meets basic requirements and it is also rational one. The final conclusion of the paper is well specified. The paper is suitable for publishing in its current form. 

Author Response

We thank this reviewer for their positive review of this paper.

Round 2

Reviewer 2 Report

The present version is fine.

Author Response

Thank you for your kind support.

Reviewer 3 Report

The study is interesting. However, some expression is unclear and the focus is not clear.  Below are some comments to further improve the quality of the manuscript.

The introduction section is too general. Which gap in the literature does this study intend to fill, and what contribution does it make to the literature? Please highlight more in the introduction section.

The main objectives of this study and the novelty point should be discussed clearly and in detail. Make the novelty clearer and in the context of existing literature.

Minor editing

Author Response

1. The introduction section is too general. Which gap in the literature does this study intend to fill, and what contribution does it make to the literature? Please highlight more in the introduction section.

We have identified specific references wrt the influence of CAFOs on air quality on the LES and included them in the revised introduction to highlight the gap we are filling.

2. The main objectives of this study and the novelty point should be discussed clearly and in detail. Make the novelty clearer and in the context of existing literature.

Thank you for your persistence in pursing this clarification.  Please see page 3, line 97-109 (with changes tracked).

Round 3

Reviewer 3 Report

The article is now acceptable. 

minor edit